# Stimulation of Osteogenic Activity of Autologous Teeth Hard Tissues as Bone Augmentation Material

**DOI:** 10.3390/biology13010040

**Published:** 2024-01-11

**Authors:** Jan Kučera, František Lofaj, Zuzana Nagyová-Krchova, Natália Šurín Hudáková, Marek Vojtko, Vitěslav Březina

**Affiliations:** 1Department of Dentistry and Maxillofacial Surgery, Faculty of Medicine, Pavol Jozef Safarik University in Kosice, Tr. SNP 1, 040 01 Kosice, Slovakia; 2Institute of Materials Research, Slovak Academy of Sciences, ÚMV SAV Košice, 040 01 Kosice, Slovakia; flofaj@saske.sk (F.L.); mvojtko@saske.sk (M.V.); 3Department of Stomatology and Maxillofacial Surgery, Faculty of Medicine, Pavol Jozef Safarik University in Kosice, Tr. SNP 1, 040 11 Kosice, Slovakia; zuzananagyova00@gmail.com; 4Department of Microbiology and Immunology, University of Veterinary Medicine and Pharmacy in Kosice, Komenskeho 73, 041 81 Kosice, Slovakia; natali.hudy@gmail.com; 5Department of Stomatology, Faculty of Medicine, Masaryk University, Kamenice 5, 625 00 Brno, Czech Republic; brezinavita@gmail.com

**Keywords:** bone augmentation materials, natural autologous hard teeth tissues, osteogenic activity, demineralization

## Abstract

**Simple Summary:**

Correction of bone volume loss after tooth extractions, periodontal lesions or trauma is performed by augmentation procedures using various biomaterials used to fill bone defects. Tissue engineering searching, often using a combination of stem cells, particles and growth factors, is making significant progress in this field. An alternative is the use of autologous tooth hard tissue grit obtained from extracted teeth. During the demineralization of the grit, the organic components of the tooth tissue, the dentin matrix, which has a confirmed high osteogenic effect, is exposed. This work evaluates the stimulatory effect of some demineralization procedures on the exposure of the dentin matrix and thus on the improvement in their osteogenic properties.

**Abstract:**

The issue of bone volume loss is playing an increasing role in bone tissue engineering. Research has focused on studying the preparation and use of different types of human or xenogenic materials and their osteogenic properties. An alternative source for this purpose could be autologous extracted teeth. The simple preparation protocol, minimal immune response, and rapid organizing of the newly formed bone with optimal mechanical properties predispose autologous hard teeth tissues (HTTs) as a promising material suitable in the indication of augmentation of maxillary and mandible defects, comparable to other high-end augmentation materials. The aim of this study was to experimentally evaluate the osteogenic potential of ground native autologous HTTs prepared by different demineralization procedures, aimed at potentiating the osteoinductive and osteoconductive properties of their organic components. The results indicate that the most effective preparation process for HTT stimulation is the application of Cleanser for 10 min followed by exposure to 0.6 N HCl for 5 min with a wash in phosphate-buffered saline solution.

## 1. Introduction

The topic of bone volume loss replacement has been given great attention in the world literature. Tissue engineering [1], as a new concept in regenerative medicine, involves searching for and using a combination of stem cells, particles and growth factors many times, and is making significant progress in this field. One of the alternatives being investigated is autologous hard tooth tissues [2,3]. Dentin, representing about 75% of the tooth volume, is very suitable for this purpose. It contains 50% minerals (apatite–amorphous tricalcium phosphate and hydroxyapatite), 30% organic substances and 20% water [4]. The organic dentin matrix, synthesized by odontoblasts, is a mineralized organic component. It is composed of macromolecules typical of connective tissue, especially collagen, predominantly type 1 (90%) of a reticular nature with crystalline deposits characteristic of bone tissue. It also contains non-collagenous proteins (NCPs), and ethylenediaminetetraacetic acid (EDTA), a substance important for its osteoinductive and dentinoinductive effect [5]. Moreover, proteoglycans, glycoproteins, lipids, dentin-specific proteins (dentin phosphoprotein (DPP) and dentin sialoprotein (DSP)), dentin matrix protein (DMP1), bone sialoproteins (DSP), osteocalcin, osteopontin and osteonectin are also present. The growth factors (GFs), such as transforming growth factor (TGF-β1), insulin growth factor (IGF), bone morphogenetic protein (BMP) and angiogenic growth factor (AGF), are also important [6]. Experimental studies studying the contribution of dentin in the process of tissue regeneration highlighted the importance of organic components [7]. GFs can induce the development of dentin degeneration or, on the contrary, stimulate reparative dentinogenesis, reparative processes in dentin and osteogenesis [8,9,10,11]. These have an osteoconductive and osteoinductive effect, and an anti-inflammatory effect, and are involved in the induction of the differentiation of dental papilla cells into odontoblasts [12]. These in vivo and in vitro studies investigated the activity of dentin and various forms of modified dentin such as an autologous dentin demineralized matrix, dentin with the extraction of NCPs and denatured dentin, on osteogenesis and dentinogenesis [13,14,15,16]. There is also the possibility of using natural unmodified autologous dentin, which has been used as a comparative specimen in the evaluation of different forms of modified dentin [17]. Hyo et al. evaluated the osteogenetic potential activity of a DDM (dentin demineralized matrix) in supporting the osteogenetic activity of MG—63 cells and compared it to a mixture of inorganic materials. Immunofluorescence assays showed that the fluorescence intensities of osteocalcin and osteonectin as biomarkers of cellular differentiation were higher on the DDM than a comparative specimen [18]. Non-significant differences in osteogenic activity between contaminated ADDM and the group after sterilization using EO gas were confirmed [19].

The aims of this study include in vitro analysis of the osteogenic activity of an autologous dentin demineralized matrix (ADDM) in its native form and after application of different forms of demineralization activating the organic matrix, in order to evaluate the most effective preparation procedure for an osteogenic effect.

## 2. Materials and Methods

### 2.1. Preparation of HTT Samples

The technology of using autologous hard teeth tissues (HTT) in tissue engineering in bone defects was introduced into clinical practice in the period 2013–2014. The preparation methodology suggested by Binderman et al. in 2014 [20] is based on the grinding of extracted teeth and the extraction of a grain fraction with a 300–1200 µm size (Figure 1a). The teeth samples used in this study were actual extracted teeth, which had not been endodontically treated (without freezing or drying), and after removal of all mechanical debris, fillings, calculus or remnants of periodontal ligaments, etc. (Figure 1(b1)). They were grounded in the ‘Smart Dentin Grinder’ (‘Smart Dentin Grinder’, Kometa Bio, Fort Lee, NJ, USA) for 20 s. The desired grain size fraction was obtained by vibrational sieving with the grit size of 1200 µm (Figure 1(b2)).

Subsequently, the grit was disinfected in Cleanser (Figure 1(b3)) and washed (Figure 1(b4)) in sterile phosphate-buffered saline (PBS) for 5 min. The samples were then exposed to solutions inducing demineralization and which made the organic matrix accessible. The treatment conditions for the whole set of samples are summarized in Table 1.

The 1st sample, marked Natural, contained HTT grit, and was washed in PBS for 10 min. The 2nd sample, marked as Cleanser 5 min, contained HTT grit identically washed in PBS followed by chemical disinfection by being exposed to Cleanser solution (0.5 M NaOH and 20% alcohol) for 5 min, and then re-washed in PBS for 10 min. The 3rd sample (Cleanser 10 min) contained HTT pulp identically washed in PBS followed by exposure to chemical disinfection with Cleanser solution for 10 min, followed by washing in PBS for 5 min. The 4th sample (EDTA 3 min), after the identical application of PBS and Cleanser for 5 min and washing, was exposed to 10% EDTA for 3 min followed by washing in PBS for 5 min. For the 5th sample (EDTA 5 min), an identical procedure as for the 4th sample was applied; however, it was exposed to 10% EDTA for 5 min. In the case of 6th sample (HCl 0.6 N 3 min), an identical procedure as in sample no. 5 was used; however, 0.6 N HCl for 3 min was used instead of EDTA. The preparation of the 7th sample (HCl 0.6 N 5 min) was identical to that in sample no. 6 except a 5 min treatment period with 0.6 N HCl was applied.

### 2.2. Scanning Electron Microscopy and Energy-Dispersive Spectroscopy

The samples, after such treatments, were monitored in vitro using Scanning Electron Microscopy (SEM) (EVO MA 15, Zeiss, Jena, Germany) combined with Energy-Dispersive Spectroscopy (EDS), Raman spectroscopy and Live-cell dynamic imaging. Before the SEM/EDS observations, all samples were washed in ethyl alcohol and ethyl ether for 10 min and then coated with a thin layer of Au to prevent electrical charging. SEM provided topographical images of the grains within a wide range of magnifications. The observations were performed mostly at the accelerating voltage reduced to 10 kV to enhance visibility of the surface features after leaching. EDS (model MAX 80, Oxford Instruments, Oxford, UK) attached to the SEM was used for the qualitative (identifications of the elements) and semi-quantitative elemental analysis (approximate determination of their concentrations) to determine the chemical compositions of the grains and its changes after the corresponding demineralization [21]. The EDS was always performed at 20 kV because it was calibrated for this accelerating voltage. Gold was intentionally excluded from quantification in the EDS analyses.

### 2.3. Raman Spectroscopy

Raman spectroscopy measurements (model XploRA, Horiba Yvon Jobin, Palaiseau, France) were complimentary to the EDS results. Raman spectroscopy [22,23,24] is a method based on the inelastic scattering of an incident monochromatic light beam on a solid surface. The energy of the incident light is transferred to the vibrational and rotational states of the atoms and molecules in the surface layer of the material. The output is the Raman spectrum—dependent on the radiation intensity of the vibrational states on the wavenumber (inverse value of the wavelength given in cm^−1^). The Raman shift values (the positions of the individual peaks) are specific for each type of interatomic bonding in the material. This makes qualitative identification of the bonds that are present in the material possible. Quantitative analysis is difficult: only relative changes based on the comparison of peak intensities are feasible. The measurements in this study were performed using a laser with a wavelength of 532 nm without any filter and 1800T grit, and using averaging from 3 measurements. The peak positions were calibrated against Si.

### 2.4. Live-Cell Dynamic Imaging

The locomotion activity of MG 63 cells in a lysate was investigated in vitro on the untreated HTT grains and after treatment with Cleanser, EDTA 10% and 0.6 N HCl using a time-lapse microcinematography technique based on video-enhanced transmitted light microscopy (The phase-contrast microscope, Nikon, Tokyo, Japan) known as “Live cell dynamic imaging” [25]. Time-lapse microscopy is a method that extends live-cell imaging from a single observation in time to the observation of cellular dynamics over long periods of time. The qualitative assessment of the images assumes measurements or counting of the corresponding quantitative parameters [26]. MG 63 cells were of the osteosarcoma origin from the osteosarcoma collection (ECACC no. 86051601, American Type Culture Collection (ATCC), Rockville, MD, USA). MG-63 cells are a type of osteoblast cell line with fibroblast morphology isolated from the bone of patient with osteosarcoma. This is a hypotriploid human cell line. The cells were cultivated in a thermostat at 37 °C and an atmosphere of 5% CO_2_ in an environment with a relative humidity of 95%. Live cell dynamic imaging is based on sequential photographs in jpg format compressed into videos. The observations were performed using NIKON Biomat (Japan) for the parallel tracking of multiple fields of view from a single experimental dish. OLYMPUS IC (Tokyo, Japan) for longer-term experiments in flow-through culture or in culture with medium was used. Qualitative assessment of the images obtained by time-lapse imaging were conducted after 12, 24, 48, 72, and 120 h.

### 2.5. Statistical Analysis

Statistical analyses were carried with IBM SPSS Statistics program (IBM Corp. Released 2021. IBM SPSS Statistics for Windows, Version 28.0. Armonk, NY, USA: IBM Corp) using the non-parametric Kruskal–Wallis test for independent samples. If the result of the Kruskal–Wallis test was ‘significant’, i.e., occurrence of at least one significant difference (*p* < 0.05), Dunn’s multiple comparison post-test was performed.

## 3. Results

### 3.1. SEM Observations

SEM images in Figure 2a–g obtained at the same magnification (×5000) illustrate changes in the progressively demineralized surfaces of individual HTT scaffolds (grains) as a function of the solution used and exposure time. Demineralization treatment of HTT exposed ADDM and its organic components. The least exposed surface corresponds to the reference (Natural) sample (Figure 2a). In Figure 2b–f, the gradually decreasing detrital deposits and the progressive demineralization of the surface can be seen. Figure 2b,c shows only the absence of detritus. Figure 2d–f shows gradual exposure of organic matrix with visible dentin tubules after application of demineralization solutions. The most exposed surface was obtained in the sample treated with 0.6 N HCl for 5 min (Figure 2g). Its surface was clean, no detritus was present, wide open tubular holes were visible after treatment, and the organic matrix was completely exposed.

### 3.2. EDS Analysis

To quantitatively characterize the effects of demineralization, EDS measured concentrations of Ca, O, P and C on the dentin surfaces after the corresponding treatments were compared with the values in the reference—the untreated HTT (Natural) sample. Figure 3 shows the EDS spectrum in the dentin zone of HTT without any treatment. The peaks are attributed to Ca, P, O, Mg, Na, and C (and maybe Cl). The focus was on the concentrations of Ca and P, which are the main elements in the apatite phases. Their concentrations were around 27 wt% for Ca and approximately 12 wt% for P. The concentrations of Na, Mg and Cl were below the limit of accuracy of the EDS method and can be neglected.

The treatment by EDTA for 3 min and 5 min (see Figure 4 and Figure 5, respectively) reduced the concentrations of Ca to 18 wt% and 5 wt% and P concentrations to 6 wt% and <1 wt%, respectively. The demineralization was even stronger after the application of HCl (Figure 5). The application of Cleanser resulted in slight increases in Ca and P concentrations, most probably due to the elimination of detritus (see later comparison in Figure 6). The Ca concentration after 5 min was only 2.5 wt% and P was completely eliminated. The process of demineralization and the exposure of the organic matrix was simultaneously accompanied by increases in C and O concentrations, which enhanced its osteoinductive and osteogenetic activities.

EDS analysis evaluated the Ca and P values of the HTT particles surfaces and was performed at different locations on scaffolds surfaces, and showed differences between groups. EDS analysis confirmed a progressive etching of dentin surfaces in HTT accompanied by a decrease in P and Ca due to their leaching when using EDTA and HCl solutions. A minor demineralization effect was observed with 10% EDTA versus 0.6 N HCl. EDS spectra from the dentin region after 3 and 5 min of exposure in the HCl solution showed that the minimum levels of Ca and P peaks were completely eliminated.

### 3.3. Raman Spectroscopy

Figure 7 shows the reference Raman spectra attributed to the (PO_4_)^−3^ group in the hydroxyapatite (Ca_5_(PO_4_)_3_) and tricalcium phosphate (Ca_3_(PO_4_)_2_), the main inorganic compounds in the dentin [23]. The peaks of (PO_4_)^−3^ occurred at 1077, 1049, 960, 590 and 431 cm^−1^. The medium-intensity peaks at 440 cm^−1^ and 596 cm^−1^ were attributed to the symmetric and asymmetric bending of (PO_4_)^−3^, respectively. The most intensive peak at 960 cm^−1^ was associated with the symmetric stretching of the “free” tetrahedral (PO_4_)^−3^ ion [24]. Two small satellite peaks at 1077 cm^−1^ and 1049 cm^−1^ corresponded to the asymmetric stretching vibrations of (PO_4_)^−3^. This reference spectrum was used to identify the changes in the apatite phases in dentin after treatment in different solutions.

Figure 8 compares the Raman spectra from the natural HTT and after its treatment by Cleanser for 5 and 10 min, respectively. Despite lower intensities, the spectra contained all peaks corresponding to the (PO_4_)^−3^ groups in Figure 7 and some additional peaks around 890 cm^−1^ and 780 cm^−1^, and wide band above 1250 cm^−1^. The increase in the intensities of the (PO_4_)^−3^ peaks (at 960, 590 and 430 cm^−1^) after the application of Cleanser for 5 min indicates that it was exposed by this treatment. This spectrum was subsequently used as a reference to reveal the effects of treatment in EDTA and HCl.

Figure 9 illustrates the changes induced by subsequent leaching in EDTA for 3 and 5 min. The cleanser 5 min sample was compared to the Cleanser 10 min sample. The relative intensities of the main (960 cm^−1^) and secondary peaks of the (PO_4_)^−3^ group remained approximately the same, which implied only a weak activation of the phosphate group. The largest changes in the background intensity occurred in the range above 1000 cm^−1^. The reason for such an effect was not clear.

The Raman spectra obtained from the dentin area of HTT after treatment in HCl were principally the same but the differences in the relative intensities after treatment were larger, and the background above 1000 cm^−1^ was absent (Figure 10).

Interestingly, the highest intensities of the (PO_4_)^−3^ peaks were observed after 3 min of treatment; after 5 min treatment, the intensity of the 960 cm^−1^ peak was reduced, most probably due to the removal of P and Ca from the top zone of the dentin surface.

### 3.4. Live-Cell Dynamic Imaging

The locomotion activity of the MG 63 cells in grit HTT lysate after exposure to 0.6 N HCl was observed. The MG 63 cells migrated to the scaffold and concentrated on its surface (see Figure 11, Figure 12 and Figure 13). A large proportion of the scaffolds in the culture exhibited this property. The migration rate of the cells to the grain corresponded to approximately 60 µm/h. (Data by time-lapse imaging after 12, 24, 48, 72, and 120 h).

### 3.5. Statistical Analysis

Statistical analysis was performed for the EDS. Differences in the levels of the of values Ca and P between the groups were explored with the Kruskal–Wallis non-parametric analysis of variance followed by the Dunn’s multiple comparison post-test for all pairwise comparisons.

The bar graphs (Figure 14) show the mean measured values with variance and the standard deviation. The arithmetic means of the measured values are shown in Figure 6. The bars (Figure 14 and Figure 15) represent the mean ± standard deviation of the measured Ca and P values for each group evaluated. EDS spectra from the dentin region after 3 and 5 min of exposure in HCl solution show that the minimum level of Ca and P peaks was completely eliminated. Therefore, the statistical evaluation of P after 3 and 5 min of treatment in HCl was not evaluated. The result of the Kruskal–Wallis test was statistically “significant” (*p* < 0.001). Dunn’s multiple comparison posttest was performed. Statistically significant results for the evaluated groups were found.

## 4. Discussion

The osteoinductive potential of dentin was reported as early as 1967, and since then many authors have focused on the issue [27]. The HTT can serve very successfully as a bone graft because many components of alveolar bone and dental tissues have an identical ontogenetic basis and histological studies confirm a significant osteoinductive effect of autologous grit, especially when fresh HTT is used [28]. The histograms confirmed bone and blood vessel formation around the dentin corpuscles. It does not induce any immune response [29,30]. Particles of dentin have a high osteoconductive ability, are not subject to resorption, are progressively enveloped by the newly miniaturized bone, the present dentin tubules, are opened by disinfection, and further improve the connection and overgrowth with the bone [31,32,33]. The ankylotic connection of bone and particles occurs. Although the enamel, which is part of the grits, has no osteogenic effect, it also ankylotically connects with the newly formed bone, which enhances densification [34,35]. Clinical studies based on histological, radiological and immunological examinations have presented the formation of a dentin–bone complex after 24 weeks. The DDM plays an important function for the growth of new bone and its connection with the surrounding bone and cement and the surface of the enosseous implant, by the cell spreading of gingival fibroblasts and their proliferation by increasing cell growth, proliferation and synthesis of the extracellular matrix [36].

The significance of the HTT granule size which we used for the analysis (300–1200 µm) is in agreement with the findings of other authors. A number of papers have stressed that the optimum pore size of a scaffold must be larger than 300 µm to allow invasion by vascular sprouts and osteoprogenitor cells [37,38].

The SEM observations (Figure 3, Figure 4 and Figure 5) confirmed that HTT treated by grinding showed moderate heterogeneity between the individual particles. The EDS method is based on the analysis of the energy spectra emitted by the electrons from the individual atoms during their return from the excited to base state. The excitation is provided by the incident electron beam of the SEM. The energies of such transitions are characteristic for each element and the intensities of the corresponding peaks are proportional to the concentration of the given element in the material. EDS is therefore widely used for the qualitative and semi-quantitative elemental analysis of the inorganic materials [39]. However, it can be successfully used also in organic materials which are not sensitive to the damage by an electron beam, as it is in HDT. Despite the limited accuracy of EDS in quantitative measurements, the observed changes in the concentrations of some elements clearly demonstrate the effects of the different treatments. The comparison of these concentrations with the results of a live-cell-imaging cell can be therefore used to elucidate the effects of these elements on the segregation and proliferation of the MG63 cells and effectiveness of the studied treatments.

The EDS spectrum and composition taken from the dentin region of HTT without treatment showed high levels of O, C, Ca, P and partial Na and Cl ions. This shows the presence of hydroxyapatite and tricalcium phosphate on the surface as well as the presence of organic components found in detritus.

SEM observations and EDS analysis confirmed a progressive etching of dentin surfaces in HTT accompanied by a decrease in P and Ca due to their leaching when using EDTA and HCl solutions. The simultaneous increase in C and O concentrations indicated an increase in the amount of organic components on the surface. The simultaneous increase in C and O concentrations indicated an increase in the amount of organic components on the surface. The decreases in the levels of P and Ca ions, which are part of hydroxyapatite (Ca_5_(PO_4_)_3_) and tricalcium phosphate (Ca_3_(PO_4_)_2_), were due to the binding of Ca and P ions to HCl molecules through the exchange of the H ion with EDTA molecules, respectively, which are chelating agents, forming chelating bonds, causing them to drop on the grain surface. The progressive decrease in Ca P ions is proportional to the length of exposure on the surface. There is exposure of the organic matrix and thus a rise in the concentration of C and O on the surface of the scaffolds. Thus, there is an increase in the osteogenic activity of autologous dentin.

Raman spectroscopy is an optical technique capable of obtaining the biochemical information of a sample in situ [40]. The spectra of both calcium hydroxyapatite and tricalcium phosphate have been obtained and chemically identified.

Raman spectroscopy in natural HTT showed strong contamination of the scaffold surface with organic detritus (Figure 8). The application of the Cleanser exposed the mineral components of dentin, which was reflected by an increase in the Raman intensities of the peaks characteristic for the (PO_4_)^−3^ group in the hydroxyapatite. The process was time-dependent, as indicated by higher intensities after testing for a longer time.

Surface treatment with 10% EDTA and 0.6 N HCl caused the strong leaching of Ca^+2^ and P ions. Despite this, the intensities of the Raman spectra of the (PO_4_)^−3^ group remained approximately the same or even slightly higher than in the reference dentin subjected to the treatment by Cleanser (see Figure 9 and Figure 10). The effect was most visible after exposure to 0.6 N HCl for 3 min, which is when Ca was already completely leached away (see Figure 6). The simultaneous presence of P indicated that Ca ions leached much faster, resulting in the increase in the relative amount of the (PO_4_)^−3^ group. This could also explain the controversy between the absence of Ca, the reduction in the P concentration and the increase in the (PO_4_)^−3^ intensities in the Raman peaks in this case. The additional peaks were not identified but their presence suggests that the dentin consisted not only from the hydroxyapatite and tricalcium phosphate but also contained organic compounds, possibly collagen III and collagen I. 

Live cell dynamic imaging visualized in vitro the dynamics of the mitotic growth and migration of MG63 osteosarcoma cells under study. MG-63 cells can be seeded onto various surfaces, such as Bioglass disks, titanium (Ti-6Al-4V) disks, and HTT scaffolds, making them a commonly used osteoblastic model to study bone cell viability, adhesion, and proliferation. When MG-63 cells are exposed to transforming growth factor β (TGFB) for example, alkaline phosphatase activity can only be induced significantly above basal levels, demonstrating the potential of MG-63 cells when studying the effects of bone differentiation. Compared to other osteosarcoma cell lines, MG-63 cells proliferate rapidly, and their increased alkaline phosphatase activity renders them suitable to study human osteoblast-like cells with regard to the regulation and expression of osteocalcin. (General ATCC producer information, MG-63 product description). Organic matrix components, mainly NCP but also other components such as TGFB and osteocalcin, pass into the granular lysate of the HTT and, depending on the accessibility of the organic matrix by demineralization, accelerate the growth and mitotic activity of MG 63 cells, which subsequently form cell colonies and bridges between the scaffolds.

MG 63 represented rapidly dividing bone cells, even with their affinity for autologous HTT. Most of the scaffolds act on the cells in a stimulatory manner in terms of migration, and the random movement of the experimental MG 63 cells is converted into chemotactic movement. The differences in MG 63 cell accumulation in natural HTTs were due to the fact that all healthy dental tissues, both dentin and enamel, were used for the experiment, which have a significantly higher mineral content and thus low osteogenic activity. When the particles are stimulated, locomotion and the formation of bridges between the skeletons occurs after 3–5 days. The results of dynamic live cell imaging were also in agreement with other presented findings evaluating cell growth by time-lapse methods, e.g., the positive effect of HTT on the growth of gingival fibroblasts on the implant surface and on their ability to produce collagen [41]. Ingendoh-Tsakmakidis also used the non-invasive optical monitoring of the response of living cells in a 3D model of a peri-implant in vitro and the response of tissue cells to biofilm growth along the implant [42]. These works fully agree with the increase in the locomotion activity of the MG 63 cells in the HTT lysate observed in the current study after exposure to 0.6 N HCl and 10% EDTA. The proportional migration of cells was not observed.

Figure 11 shows that the migration occurred only on some parts of particles. The reason seems to be the consequence of local contamination by detritus and/or the presence of enamel, which blocked the possibility of the osteogenic activity of the dentin matrix. The areas without detritus after grinding showed an increase in osteogenic activity, manifested by a higher concentration of migrating cells. The enamel did not allow exposure of the dentin matrix and, thus, reduced the migratory effect. Figure 13 demonstrates the ultimate result of the increase in osteogenic activity after leaching via the formation of organized cross-links between the exposed ADDM particles. This enhancement was related to the demineralization via the removal of Ca and P ions from the HTT grain surface.

Statistical EDS analysis was performed. Raman spectroscopy and live cell imaging both provided qualitative data and were not statistically evaluated. Statistical evaluation of the EDS confirmed significant differences for different HTT stimulations. The zero level of the P values after 0.6 N HCl stimulation (Figure 15) indicates the P ion efflux from the scaffold surface. The significant EDS differences confirm the use of HCl as the most effective stimulation modality among the threat-management procedures used in this work.

Thus, the aim of this work that for the verification of the osteogenic activity of an autologous dentin matrix activated by demineralization to be experimentally confirmed. The use of demineralized fresh, topically extracted, unfrozen, or undried HTTs induced osteogenic processes by applying their organic components in the processes in question. The inclusion of demineralization and the removal of Ca and P ions from the HTT grain surface in the process preparation protocol enhanced the increase in osteogenic activity and can ensure faster bone completion in alveolar ridge preservation techniques in clinical practice.

## 5. Conclusions

SEM/EDS and Raman spectroscopy investigation in the current study showed that
Leaching of Ca and P in the EDTA and HCl solutions in the given concentration from the surface of HTT grains is proportional to the exposure time;Leaching treatment exposes organic components in the autologous dentin matrix by means of the inorganic hydroxyapatite part of the dentin;The increase in organic components increases the osteogenic activity of autologous dentin;The most effective HTT stimulation seems to be the application of Cleanser for 10 min followed by exposure to 0.6 N HCl for 5 min while applying a wash in PBS after each step of the preparation protocol.


The experimental results confirmed the validity of the use of autologous HTT.

## Figures and Tables

**Figure 1 biology-13-00040-f001:**
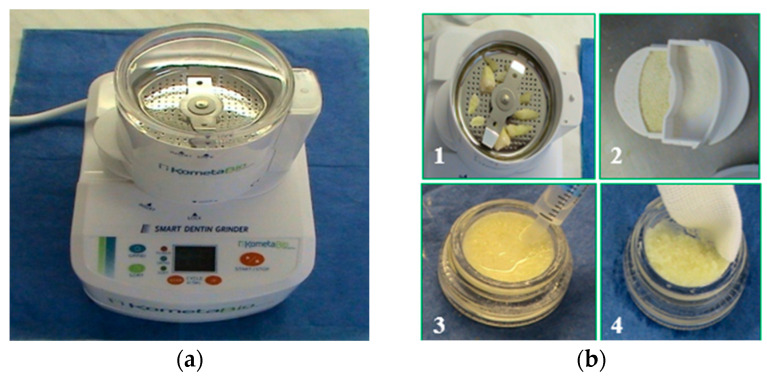
Preparation procedure of hard tooth tissues (HTT). (**a**) ‘Smart Dentin Grinder’ used for natural teeth grinding. (**b**) Preparation steps: (**1**)—mechanically cleaned teeth prepared for grinding; (**2**)—HTT grafting material after griding and sieving with the size of 300–1200 µm; (**3**)—disinfection in the Cleanser solution for 5 min; (**4**)—washing in sterile phosphate-buffered saline (PBS) for 5 min.

**Figure 2 biology-13-00040-f002:**
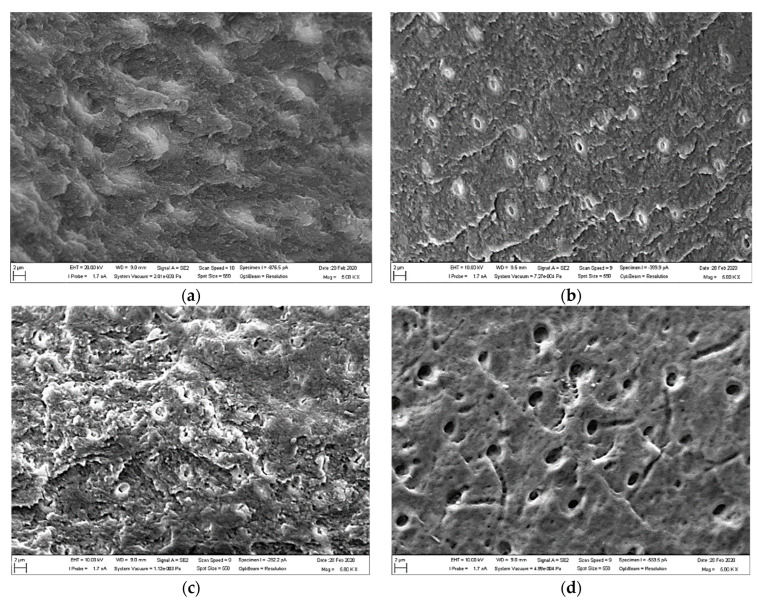
SEM images of the demineralization treatment on HTT topography in the samples: (**a**) Natural (natural HTT), (**b**) Cleanser 5 min, (**c**) Cleanser 10 min, (**d**) EDTA 3 min, (**e**) EDTA 5 min, (**f**) HCl 0.6 N 3 min, and (**g**) HCl 0.6 N 5 min.

**Figure 3 biology-13-00040-f003:**
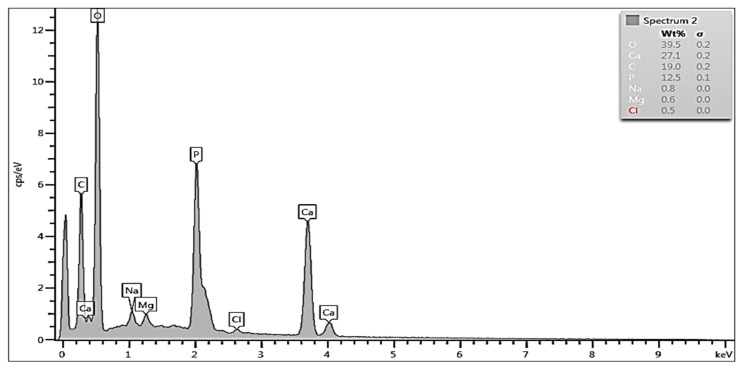
EDS spectrum and composition taken from the dentin area of HTT without any treatment.

**Figure 4 biology-13-00040-f004:**
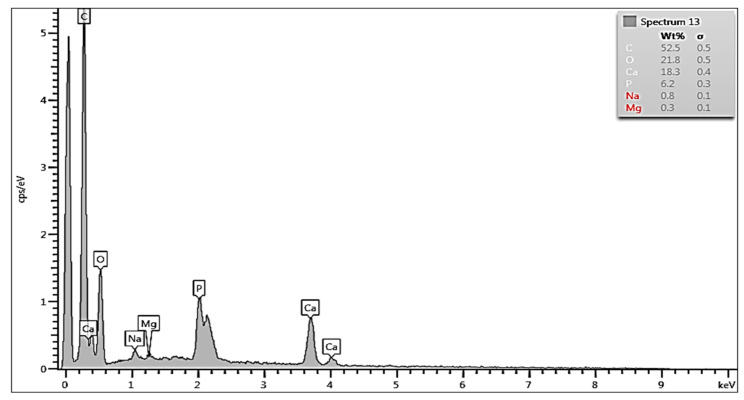
EDS spectrum of the dentin area of HTT after treatment with 10% EDTA for 3 min. Leaching of Ca and P is indicated by the reduction in the intensities of the corresponding peaks.

**Figure 5 biology-13-00040-f005:**
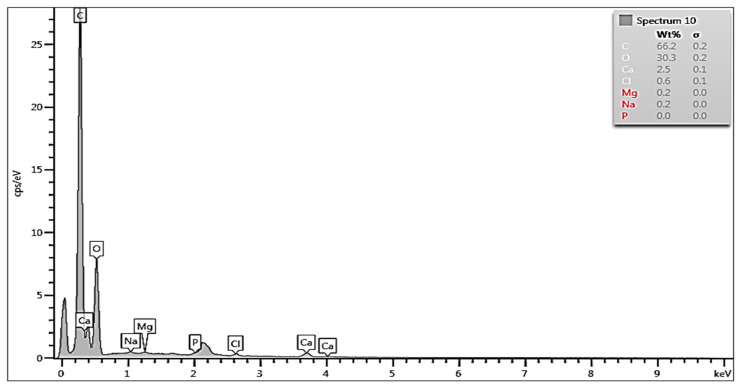
EDS spectrum from the dentin area after 5 min of treatment in HCl solution. Note that the peaks of Ca were minimally and P were completely eliminated.

**Figure 6 biology-13-00040-f006:**
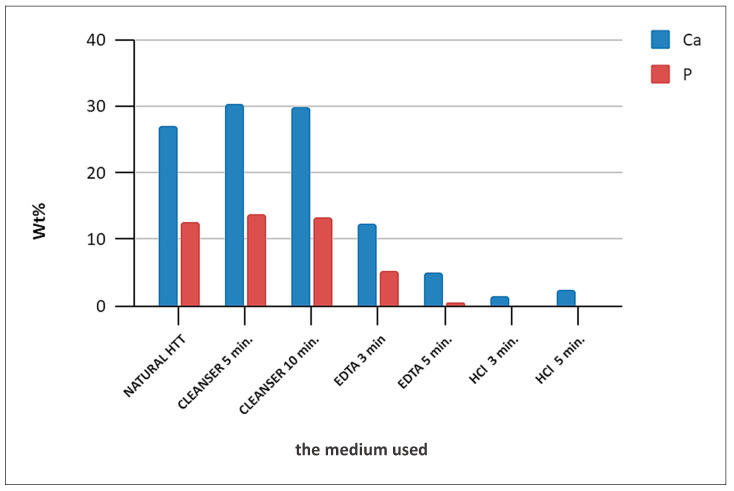
Summary of the changes of Ca and P concentrations (the arithmetic means of the measured values) in the dentin area of HTT after demineralization using the Cleanser, EDTA and HCl solutions.

**Figure 7 biology-13-00040-f007:**
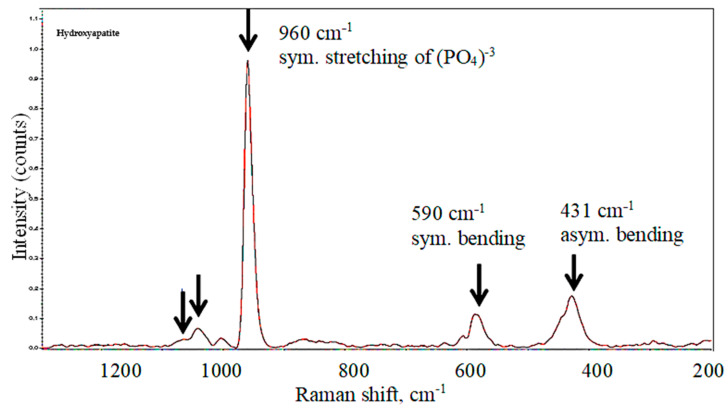
Reference Raman spectrum of (PO_4_)^−3^ group in the hydroxyapatite and tricalcium phosphate [22,23].

**Figure 8 biology-13-00040-f008:**
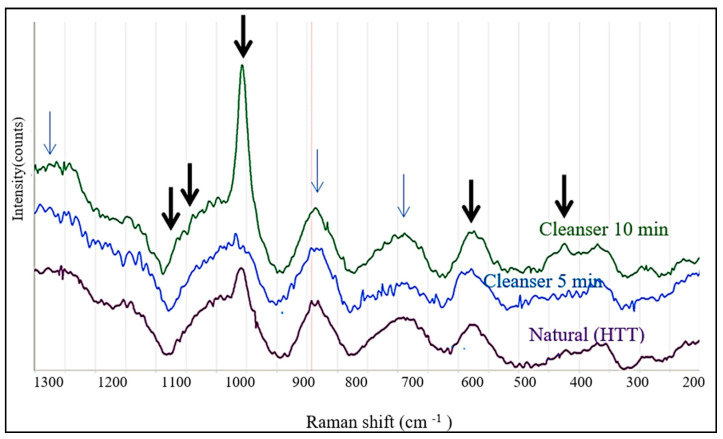
Comparison of Raman spectra in natural HTT, and after stimulation by Cleanser for 5 min and for 10 min. The thick arrows indicate the peaks of (PO_4_)^−3^, and thin arrows indicate unidentified peaks.

**Figure 9 biology-13-00040-f009:**
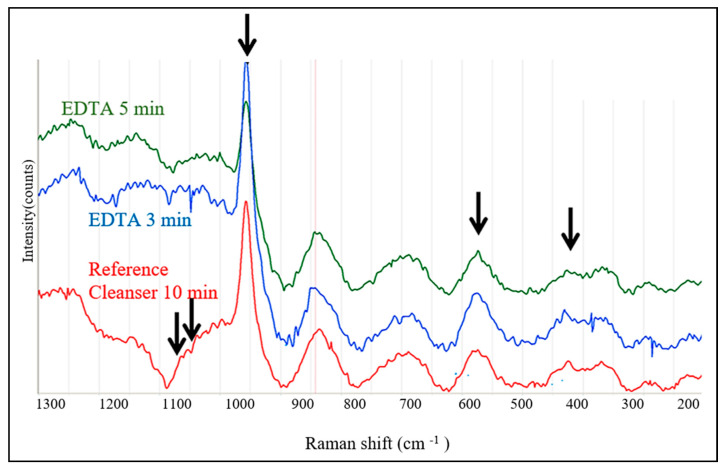
Comparison of Raman spectra after treatment by EDTA for 3 min and 5 min compared to the reference sample, HTT, stimulated by Cleanser for 10 min. The arrows indicate (PO_4_)^−3^ peaks.

**Figure 10 biology-13-00040-f010:**
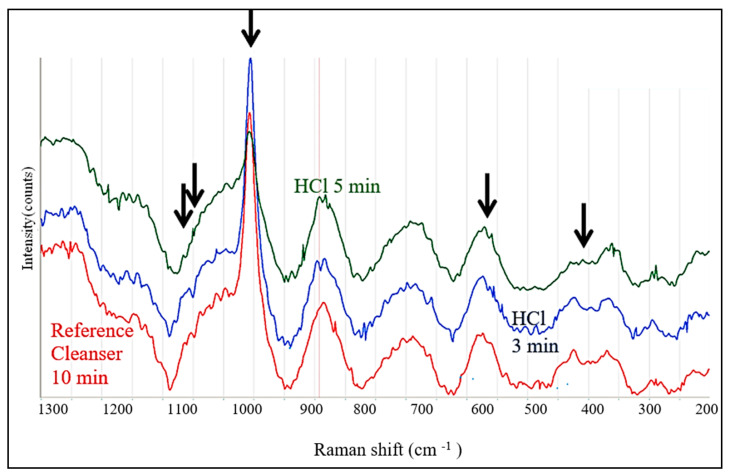
The changes in Raman spectra after treatment in HCl 0.6 N for 3 min and 5 min in comparison with the reference HTT sample. The arrows indicate (PO_4_)^−3^ peaks.

**Figure 11 biology-13-00040-f011:**
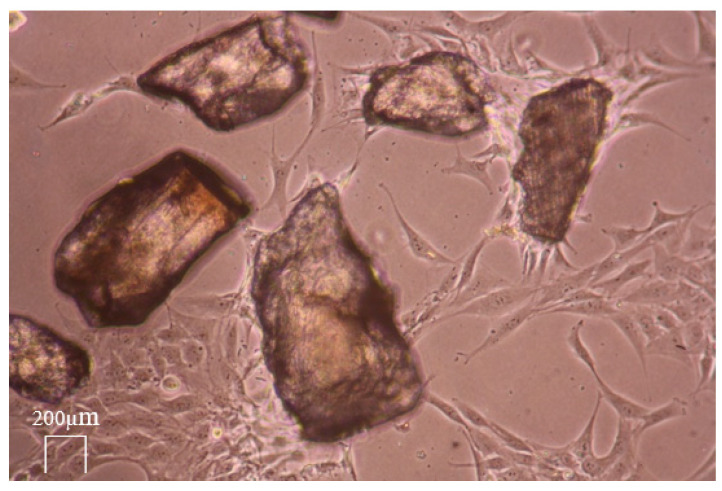
The untreated HTT inoculated in MG 63 cell culture with medium. Live cell dynamic imaging on sequential photographs after 24 h presents the affinity of the cells to only some scaffolds.

**Figure 12 biology-13-00040-f012:**
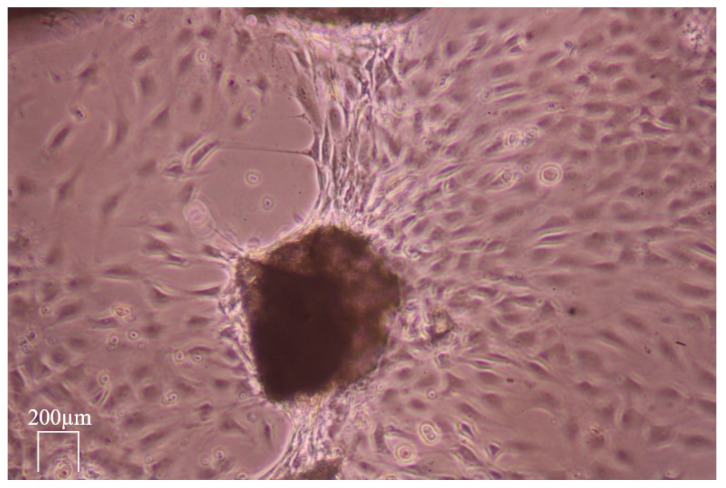
The ADDM particles stimulated by treatment with 0.6 N HCl. Live cell dynamic imaging on sequential photographs after 72 h presents the differences in MG 63 cell accumulation in some areas of ADDM particles.

**Figure 13 biology-13-00040-f013:**
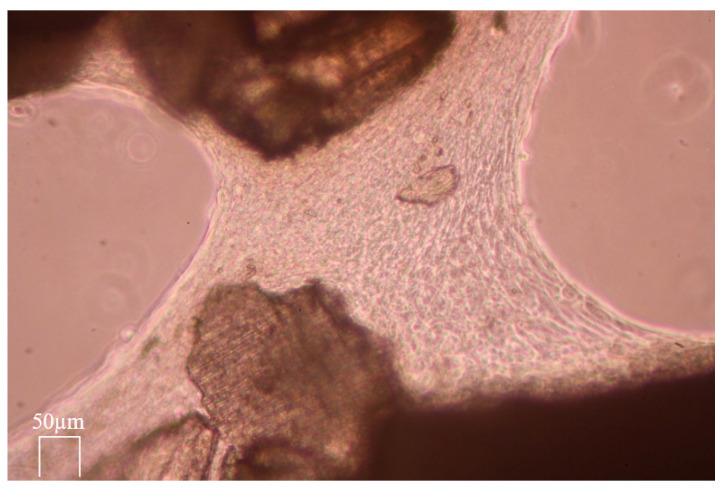
Formation of cell crosslinks MG 63 after exposure to 10% EDTA forming an organized layer of cells. Live cell dynamic imaging on sequential photographs after 5 days.

**Figure 14 biology-13-00040-f014:**
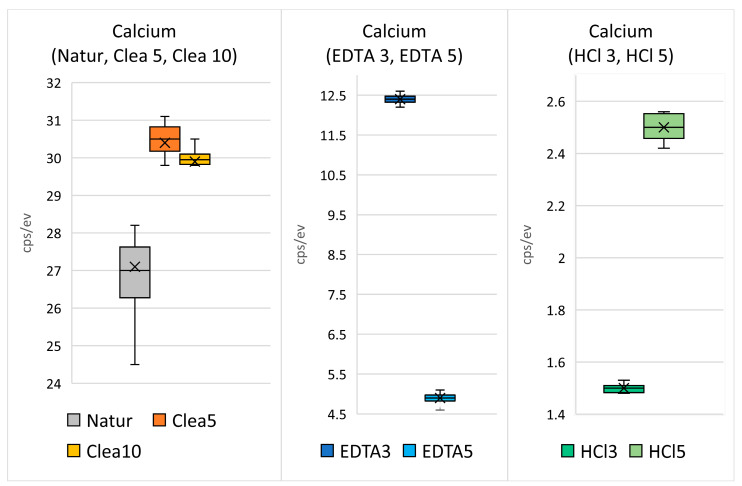
Statistical analysis of the level of Ca concentration values in the dentin area of the untreated HTT and after demineralization with the Cleanser, EDTA and HCl solutions. (Natur—untreated HTT—the 1st sample), Clea5 (HTT treated by chemical disinfection exposure to Cleanser solution for 5 min—the 2nd sample), Clea10 (HTT treated by chemical disinfection exposure to Cleanser solution for 10 min—the 3rd sample), EDTA 3 (HTT treated by exposure to EDTA for 3 min—the 4th sample), EDTA 5 (HTT treated by exposure to EDTA for 5 min—the 5th sample), HCl3 (HTT treated by exposure to HCl for 3 min—the 6th sample), and HCl5 (HTT treated by exposure to HCl for 5 min—the 7th sample).

**Figure 15 biology-13-00040-f015:**
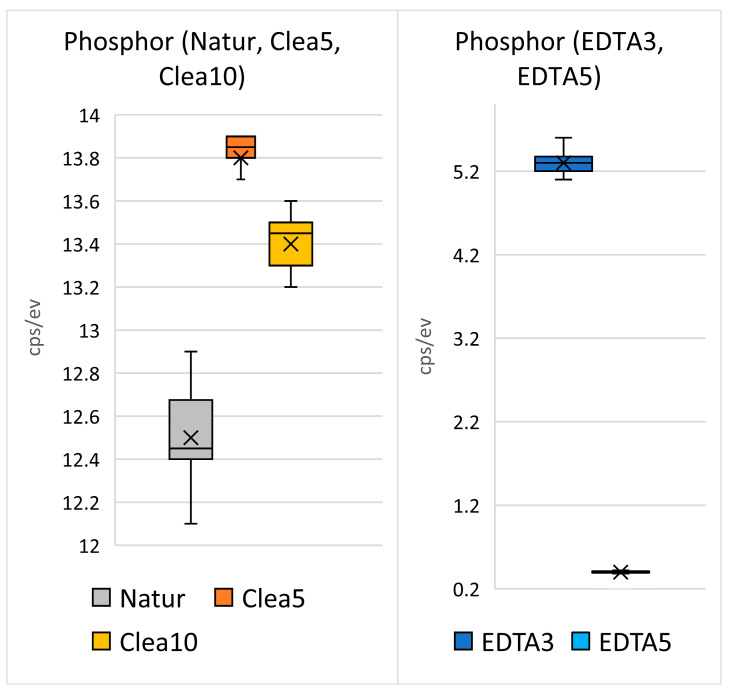
Statistical analysis of the levels of P concentration values in the dentin area of the untreated HTT and after the demineralization with Cleanser and EDTA solutions. Statistical evaluation of P after 3 and 5 min treatment in HCl was not evaluated because zeroth P values were observed after HCl stimulation (Figure 6). (Natur—untreated HTT—the 1st sample), Clea5 (HTT treated by chemical disinfection exposure to Cleanser solution for 5 min—the 2nd sample), Clea10 (HTT treated by chemical disinfection exposure to Cleanser solution for 10 min—the 3rd sample), EDTA 3 (HTT treated by exposure to EDTA for 3 min—the 4th sample), and EDTA 5 (HTT treated by exposure to EDTA for 5 min—the 5th sample).

**Table 1 biology-13-00040-t001:** The treatment parameters of the individual HTT samples for analysis.

	Washing (PBS)	Cleanser	Washing (PBS)	EDTA 10%	HCl 0.6 N	Washing (PBS)
Natural	10 min					
Cleanser 5 min	10 min	5 min	5 min			
Cleanser 10 min	10 min	10 min	5 min			
EDTA 10% 3 min	10 min	5 min	5 min	3 min		10 min
EDTA 10% 5 min	10 min	5 min	5 min	5 min		10 min
HCl 0.6 N 3 min	10 min	5 min	5 min		3 min	10 min
HCl 0.6 N 5 min	10 min	5 min	5 min		5 min	10 min

EDTA—10% solution of the ethylenediaminetetraacetic acid, HCl—0.6 N solution of the hydrochloric acid, Cleanser—0.5 M of NaOH and 20% alcohol (*v*/*v*), PBS—sterile phosphate-buffered saline.

## Data Availability

Data are contained within the article.

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
