# Peer review of "Stimulation of Osteogenic Activity of Autologous Teeth Hard Tissues as Bone Augmentation Material"

_biology, 2024, doi:10.3390/biology13010040_

Round 1
Reviewer 1 Report
Comments and Suggestions for Authors
I express my gratitude to the authors for conducting the study that assessed a multitude of parameters. Nevertheless, I believe that a number of the elements should be reorganized.
Why did you use ''MG 63 cells'' for Live-cell dynamic imaging? Wouldn't it be more appropriate to use healthy bone stem cells instead of cancer cells to reflect the clinic results? In addition, it would be more appropriate to use dental pulp-derived stem cells to evaluate dentinogenic processes.
Kindly include the country and company of origin for any products attributed to a private company that are referenced in the manuscript.
Discuss the results of EDX analysis in the discussion section. What could be the cause of ion exchanges? What is the relationship of the differences in ion exchange with the other parameters you evaluated? Kindly explain the potential clinical consequences of this situation.
Although many parameters were evaluated in the study, the discussion part is quite short. Please discuss each of your results in detail with reference articles. Kindly incorporate human and animal study results that are pertinent to the subject of your study in the discussion
Please rearrange of the some cited studies reference style as recommended by the journal author guideline. Some of your references have incorrect source citation. Make sure that the sources you add are up-to-date. When discussing on your findings, consult sources that have been published within the past decade. Do not use outdated sources unless necessary. This way you can compare your results with current literature.
Rearrange the Author Contributions, Funding, Institutional Review Board Statement, Informed Consent Statement, Acknowledgments sections correctly.
Author Response
.

Reviewer 2 Report
Comments and Suggestions for Authors
1. The introduction is too short! A small number of authors are used and the information from the cited articles is scarce.
2. Complex terminology implies the use of abbreviations. So placed in the text make it difficult for the readers. Please list all abbreviations in a table located before the references.
3. The material is described in detail, illustrated with photos and a table. Information on the statistical methods used is missing. Please add them as a final point.
4. The results are described in detail and very well visualized through figures, graphs and charts.
5. The discussion is insufficient and poorly structured - I can't find any authors cited! In the discussion, articles with similar studies are selected and the results are compared, confirming or denying them. Please find relevant articles and discuss the results!
6. Conclusions are systematized and highlight the results.
7. Author Contributions, - you have redundant text that needs to be removed.
8. Funding, Institutional Review Board Statement, Informed Consent Statement, Data Availability Statement, Acknowledgments are not aligned with the article, but are left in form from the template. Please rewrite according to the specific article.
Author Response
.

Reviewer 3 Report
Comments and Suggestions for Authors
This work evaluates the osteogenic potential of ground native autologous HTT prepared by different demineralization procedures. The authors showed enough information to support their idea.
Here are my comments.
In table 1, the ethyl-alcohol and ethyl ether treatments are listed without explanation. In 2.1 Preparation of HTT samples, these two treatments are not introduced like other treatments in Table 1, like cleanser, EDTA, HCL… However, in 2.2 SEM and EDS section, these two treatments were introduced as the preparation for SEM/EDS characterization. It is not clear whether the ethyl-alcohol and ethyl ether treatments are for all sample preparations or just for SEM/EDS. If those are for SEM/EDS only, I suggest the authors remove them from Table 1.
In Figure 2a, natural sample was characterized with ETH=20kV, however, other samples were with EHT =10kV (Figure 2b-2g). Why did the authors use a higher voltage for the natural sample?
In Line 87, “Subsequently, the grit was disinfected in Cleanser (Figure 1 b3) and washed (Figure 1 b4) in sterile phosphate buffered saline (PBS) for 5 minutes.” The first figure introduced in the text is Figure 1 b3. I suggested the authors mention Figure 1 a, b1 and b2 somewhere in Section 2.1, as all figures should be introduced or discussed.
Author Response
.

Round 2
Reviewer 1 Report
Comments and Suggestions for Authors
Dear Author,
Thank you for your edit. Nonetheless, I believe there are a few aspects that require correction.
"The method of using MG 63 cells and allows for faster visualization of cell division and their locomotion activity." Based on your statement, I understand why you use MG 63 cells. However, in your results, you mentioned that the dentinogenic process was induced in your experimental results with these cell line. However, the MG 63 cells you use have fibroblast morphology and are related to bone tissue. How did you come up with the idea that it induces the formation of dentin tissue? For this reason, I previously mentioned that the use of dental pulp-derived stem cells may reflect dentinogenic processes.
It is possible to offer commentary on the osteogeic process as a consequence of the experiment conducted on the cell line employed.
Instead of "autologous hard teeth tissues" in the title, the phrase "autologous teeth hard tissues" may be more appropriate.
Please write your EDS research results under the heading "3.2. EDS analysis". (See; the paragraph at the end of the Result section)
Author Response
.

Reviewer 2 Report
Comments and Suggestions for Authors
Good luck!
Author Response
Thank you very much for taking the time to new review this manuscript.
Round 3
Reviewer 1 Report
Comments and Suggestions for Authors
Dear Author,
I see that you have corrected the incorrect statement you previously wrote about dentinogenesis. However, I will have an additional suggestion after your edits in the discussion section.
Please avoid repetitive mg 63 cell line definitions in the discussion section. Similar definitions have been repeated
Author Response
.
